# Playing the lottery with rewards and multiple languages: lottery tickets in RL and NLP

**Haonan Yu,**[*] **Sergey Edunov, Yuandong Tian, and Ari S. Morcos**[†]
Facebook AI Research
haonanu@gmail.com, {edunov,yuandong,arimorcos}@fb.com

## Abstract

The lottery ticket hypothesis proposes that over-parameterization of deep neural networks (DNNs) aids training by increasing the probability of a "lucky" sub-network initialization being present rather than by helping the optimization process (Frankle & Carbin, 2019). Intriguingly, this phenomenon suggests that initialization strategies for DNNs can be improved substantially, but the lottery ticket hypothesis has only previously been tested in the context of supervised learning for natural image tasks. Here, we evaluate whether "winning ticket" initializations exist in two different domains: natural language processing (NLP) and reinforcement learning (RL). For NLP, we examined both recurrent LSTM models and large-scale Transformer models (Vaswani et al., 2017). For RL, we analyzed a number of discrete-action space tasks, including both classic control and pixel control. Consistent with work in supervised image classification, we confirm that winning ticket initializations generally outperform parameter-matched random initializations, even at extreme pruning rates for both NLP and RL. Notably, we are able to find winning ticket initializations for Transformers which enable models one-third the size to achieve nearly equivalent performance. Together, these results suggest that the lottery ticket hypothesis is not restricted to supervised learning of natural images, but rather represents a broader phenomenon in DNNs.

## 1 Introduction

The lottery ticket phenomenon (Frankle & Carbin, 2019; Frankle et al., 2019; Zhou et al., 2019) occurs when small, sparse sub-networks can be found in over-parameterized deep neural networks (DNNs) which, when trained in isolation, can achieve similar or even greater performance than the original, highly over-parameterized network. This phenomenon suggests that over-parameterization in DNN training is beneficial primarily due to proper initialization rather than regularization during the training process itself (Allen-Zhu et al., 2019; 2018; Du & Lee, 2018; Du et al., 2019; Neyshabur et al., 2014; 2019).

However, despite extensive experiments in Frankle & Carbin (2019) and Frankle et al. (2019), it remains unclear whether the lottery ticket phenomenon is an intrinsic feature of DNN behavior or whether it is dependent on other factors such as supervised learning, network architecture, specific tasks (e.g., image classification), the bias in the dataset, or artifacts from the optimization algorithm itself. As discussed in Frankle & Carbin (2019) and Liu et al. (2019), large learning rates severely damage the lottery ticket effect, and for larger models (such as VGG and ResNets) and datasets (e.g., ImageNet), heuristics like learning rate warmup (Frankle & Carbin, 2019) and late rewinding (Frankle et al., 2019) are needed to induce high performance and reliable winning tickets. Recent work has also questioned the effectiveness of the lottery ticket hypothesis, raising concerns about the generality of this phenomenon (Liu et al., 2019; Gale et al., 2019).

In this work, we address the question of whether the lottery ticket phenomenon is merely an artifact of supervised image classification with feed-forward convolutional neural networks, or whether

---

[*]Work done while at Facebook AI Research; currently at Horizon Robotics
[†]To whom correspondence should be addressed.

this phenomenon generalizes to other domains, architectural paradigms, and learning regimes (e.g., environments with reward signals). Many natural language processing (NLP) models feature complex gating mechanics paired with recurrent dynamics, either of which may significantly impact the optimization process and, consequently, the lottery ticket phenomenon. Furthermore, prior work has suggested that this phenomenon is not present in Transformer models (Gale et al., 2019), calling the broad applicability of lottery tickets into question. In reinforcement learning (RL), the data distribution shifts as the agent learns from often reward signals, significantly modifying the optimization process and the resultant networks. Pre-trained feature extractors have proven successful in computer vision (Kornblith et al., 2019; Razavian et al., 2014; Yosinski et al., 2014), but in RL, agents often fail to generalize even to extremely similar situations (Raghu et al., 2018; Lanctot et al., 2017; Cobbe et al., 2018; Ruderman et al., 2019).

To answer this question, we evaluate whether the lottery ticket hypothesis holds for NLP and RL, both of which are drastically different from traditional supervised image classification. To demonstrate the lottery ticket phenomenon, we ask whether sparsified subnetworks initialized as winning tickets outperform randomly initialized subnetworks at convergence. We note that, though desirable, we do not require that subnetworks match the performance of the full network, as originally stated in Frankle et al. (2019). We exclude this requirement because we are primarily interested in whether appropriate initialization impacts the performance of subnetwork training, consistent with the revised definition of the lottery ticket hypothesis in Frankle & Carbin (2019). For NLP, we evaluate language modelling on Wikitext-2 with LSTMs (Merity et al., 2017) and machine translation on the WMT'14 English-German translation task with Transformers (Vaswani et al., 2017). For RL, we evaluate both classic control problems and Atari games (Bellemare et al., 2013).

Perhaps surprisingly, we found that lottery tickets are present in both NLP and RL tasks. In NLP, winning tickets were present both in recurrent LSTMs trained on language modeling and in Transformers (Vaswani et al., 2017) trained on a machine translation task while in RL, we observed winning tickets in both classic control problems and Atari games (though with high variance). Notably, we are able to find masks and initializations which enable a Transformer Big model to train from scratch to achieve 99% the BLEU score of the unpruned model on the Newstest'14 machine translation task while using only a third of the parameters. Together, these results demonstrate that the lottery ticket phenomenon is a general property of deep neural networks, and highlight their potential for practical applications.

## 2 RELATED WORK

Our work is primarily inspired by the lottery ticket hypothesis, first introduced in Frankle & Carbin (2019), which argues that over-parameterized neural networks contain small, sparse sub-networks (with as few as 0.1% of the original network's parameters) which can achieve high performance when trained in isolation. Frankle et al. (2019) revised the lottery ticket hypothesis to include the notion of late rewinding, which was found to significantly improve performance for large-scale models and large-scale image classification datasets. In addition, the revised lottery ticket hypothesis relaxed the need for subnetworks to match the performance of the full network to simply exceeding the performance a randomly initialized subnetwork. For brevity, we will refer exclusively to the revised definition throughout the paper. However, both of these works solely focused on supervised image classification, leaving it unclear whether the lottery ticket phenomenon is present in other domains and learning paradigms.

Recent work (Liu et al., 2019) challenged the lottery ticket hypothesis, demonstrated that for structured pruning settings, random sub-networks were able to match winning ticket performance. Gale et al. (2019) also explored the lottery ticket hypothesis in the context of ResNets and Transformers. Notably, they found that random sub-networks could achieve similar performance to that of winning ticket networks for both model classes. However, they did not use iterative pruning or late rewinding, both of which have been found to significantly improve winning ticket performance Frankle et al. (2019); Frankle & Carbin (2019).

More broadly, pruning methods for deep neural networks have been explored extensively (Han et al., 2015). Following Frankle et al. (2019) and Frankle & Carbin (2019), we use magnitude pruning in this work, in which the smallest magnitude weights are pruned first (Han et al., 2015). To determine optimal pruning performance, Molchanov et al. (2017b) greedily prune weights to determine an

oracle ranking. Also, Ayinde et al. (2019) and Qin et al. (2019) have attempted to rank channels by redundancy and preferentially prune redundant filters, and Molchanov et al. (2017a) used variational methods to prune models. However, all of these works only evaluated these approaches for supervised image classification.

# 3 APPROACH

## 3.1 GENERATING LOTTERY TICKETS

**Pruning methods** In our experiments, we use both one-shot and iterative pruning to find winning ticket initializations. In one-shot pruning, the full network is trained to convergence, and then a given fraction of parameters are pruned, with lower magnitude parameters pruned first. To evaluate winning ticket performance, the remaining weights are reset to their initial values, and the sparsified model is retrained to convergence. However, one-shot pruning is very susceptible to noise in the pruning process, and as a result, it has widely been observed that one-shot pruning under-performs iterative pruning methods (Frankle & Carbin, 2019; Han et al., 2015; Liu et al., 2019).

In iterative pruning (Frankle & Carbin, 2019; Han et al., 2015), alternating cycles of training models from scratch and pruning are performed. At each pruning iteration, a fixed, small fraction of the remaining weights are pruned, followed by re-initialization to a winning ticket and another cycle of training and pruning. More formally, the pruning at iteration $k + 1$ is performed on the trained weights of the winning ticket found at iteration $k$. At iteration $k$ with an iterative pruning rate $p$, the fraction of weights pruned is:

$$r_k = 1 - (1 - p)^k, \ \ 1 \leq k \leq 20$$

We therefore perform iterative pruning for all our experiments unless otherwise noted, with an iterative pruning rate $p = 0.2$. For our RL experiments, we perform 20 pruning iterations. Pruning was always performed globally, such that all weights (including biases) of different layers were pooled and their magnitudes ordered for pruning. As a result, the fraction of parameters pruned across layers may vary given a total pruning fraction.

**Late rewinding** In the original incarnation of the lottery ticket hypothesis (Frankle & Carbin, 2019), winning tickets were reset to their values at initialization. However, Frankle et al. (2019) found that resetting winning tickets to their values to an iteration early in training resulted in dramatic improvements in winning ticket performance on large-scale image datasets, even when weights were reset to their values only a few iterations into training. late rewinding can therefore be defined as resetting winning tickets to their weights at iteration $j$ in training, with the original lottery ticket paradigm taking $j = 0$. Unless otherwise noted, we use late rewinding throughout, with a late rewinding value of 1 epoch used for all RL experiments. We also compare late rewinding with normal resetting for NLP in section 4.1 and on several representative RL games in section 4.2.

## 3.2 NATURAL LANGUAGE PROCESSING

To test the lottery ticket hypothesis in NLP, we use two broad model and task paradigms: two-layer LSTMs for language modeling on Wikitext-2 (Merity et al., 2017) and Transformer Base and Transformer Big models (Vaswani et al., 2017) on the WMT'14 En2De News Translation task.

**Language modeling using LSTMs** For the language modeling task, we trained an LSTM model with a hidden state size of 650. It contained a dropout layer between the two RNN layers with a dropout probability of $0.5$. The LSTM received word embeddings of size 650. For training, we used truncated Backpropagation Through Time (truncated BPTT) with a sequence length of 50. The training batch size was set to 30, and models were optimized using Adam with a learning rate of $10^{-3}$ and $\beta_1 = 0.9, \beta_2 = 0.999, \epsilon = 10^{-3}$.

As in the RL experiments, we use global iterative pruning with an iterative pruning rate of 0.2 and 20 total pruning iterations. We also employ late rewinding where the initial weights of a winning ticket were set to the weights after first epoch of training the full network. For ticket evaluation, we

trained the model for 10 epochs on the training set, after which we computed the log perplexity on the test set. We also perform two ablation studies without late rewinding and using one-shot pruning, respectively.

**Machine translation using transformers**   For the machine translation task, we use the FAIRSEQ framework[1] (Ott et al., 2019), following the setup described in (Ott et al., 2018) to train Transformer-base model on the pre-processed dataset from (Vaswani et al., 2017). We train models for 50,000 updates and apply checkpoint averaging. We report case-sensitive tokenized BLEU with `multi-bleu.pl`[2] on the Newstest 2014 [3].

### 3.3   REINFORCEMENT LEARNING

#### 3.3.1   SIMULATED ENVIRONMENTS

For our RL experiments, we use two types of discrete-action games: classic control and pixel control. For classic control, we evaluated 3 OpenAI Gym[4] environments that have vectors of real numbers as network inputs. These simple experiments mainly serve to verify whether winning tickets exist in networks that solely consist of fully-connected (FC) layers for RL problems. For pixel control, we evaluated 9 Atari (Bellemare et al., 2013) games. A summary of all the games along with the corresponding networks is provided in Table A1. Classic control games were trained using the FLARE framework[5] and Atari games were trained using the ELF framework[6] (Tian et al., 2017).

#### 3.3.2   TICKET EVALUATION

To evaluate a ticket (pruned sub-network), we train the ticket to play a game with its corresponding initial weights for $N$ epochs. Here, an epoch is defined as every $M$ game episodes or every $M$ training batches depending on the game type. At the end of training, we compute the averaged episodic reward over the last $L$ game episodes. This average reward, defined as *ticket reward*, indicates the final performance of the ticket playing a game by sampling actions from its trained policy. For each game, we plot ticket reward curves for both winning and random tickets as the fraction of weights pruned increases. To evaluate the impact of random seed on our results, we repeated the iterative pruning process three times on every game, and plot (mean ± standard deviation) for all results.

**Classic control**   All three games were trained in the FLARE framework with 32 game threads running in parallel, and each thread gets blocked every 4 time steps for training. Thus a training batch contains $32 \times 4 = 128$ time steps. Immediate rewards are divided by 100. For optimization, we use RMSprop with a learning rate of $10^{-4}$ and $\alpha = 0.99, \epsilon = 10^{-8}$.

**Pixel control**   All 9 Atari games are trained using a common ELF configuration with all RL hyperparameters being shared across games (see Table A1 for our choices of $N$ and $M$). Specifically, each game has 1024 game threads running in parallel, and each thread gets blocked every 6 time steps for training. For each training batch, the trainer samples 128 time steps from the common pool. The policy entropy cost for exploration is weighted by 0.01. We clip both immediate rewards and advantages to $[-1, +1]$. Because the training is asynchronous and off-policy, we impose an importance factor which is the ratio of action probability given by the current policy to that from the old policy. This ratio is clamped at 1.5 to stabilize training. For optimization, we use Adam with a learning rate of $10^{-3}$ and $\beta_1 = 0.9, \beta_2 = 0.999, \epsilon = 10^{-3}$.

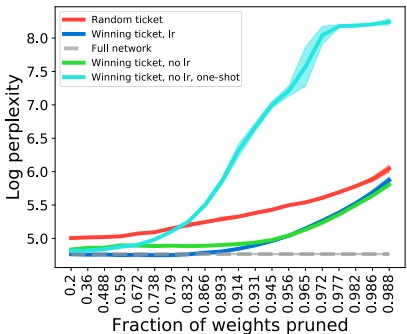

**Figure 1:** Performance of winning ticket initializations for LSTM models trained on Wikitext-2.

## 4 RESULTS

### 4.1 NATURAL LANGUAGE PROCESSING

In this section, we investigate whether winning ticket outperform random tickets in the context of NLP. In particular, we focus on language modeling with a recurrent LSTM and machine translation using two variants of the Transformer model (Vaswani et al., 2017).

#### 4.1.1 LANGUAGE MODELING WITH LSTMS

We first investigate whether winning tickets exist in a two-layer LSTM model trained to perform a language modeling task on the Wikitext-2 dataset. Encouragingly, we found that winning tickets with late rewinding significantly outperformed random tickets in this task for all pruning levels and demonstrated high performance even at high pruning rates, with as many as 90% of parameters capable of being removed without a noticeable increase in log perplexity (Figure 1).

To measure the impact of iterative pruning and late rewinding, we performed ablation studies for these two properties. Interestingly, removing late rewinding only slightly damaged performance and primarily impacted intermediate pruning levels, suggesting that it is only partially necessary for language modeling with LSTMs. Iterative pruning, however, was essential, as performance plummeted, reaching values worse than random tickets once 80% of parameters had been pruned. Together, these results both validate the lottery ticket hypothesis in language modeling with LSTMs and demonstrate the impact of iterative pruning and late rewinding in this setting.

#### 4.1.2 MACHINE TRANSLATION WITH TRANSFORMERS

We next evaluate whether winning tickets are present in Transformer models trained on machine translation. Our baseline machine translation model Transformer Base achieves a BLEU score of 27.6 on the Newstest'14 test set (compared to 27.3 in Vaswani et al. (2017)). We perform global iterative pruning with and without late rewinding to the parameters of the baseline model after 1000 updates. Consistent with our results on language modeling with LSTMs, winning tickets outperform random tickets in Transformer models (Figure 2 left). Additionally, we again found that iterative pruning and late rewinding significantly improved performance, with iterative pruning again having a larger impact than late rewinding. The necessity of these modifications explain why our results differ from Gale et al. (2019), which only used one-shot pruning without late rewinding.

---

[1]https://github.com/pytorch/fairseq
[2]https://github.com/moses-smt/mosesdecoder/blob/master/scripts/generic/multi-bleu.perl
[3]https://www.statmt.org/wmt14/translation-task.html
[4]https://gym.openai.com/
[5]https://github.com/idlrl/flare
[6]https://github.com/pytorch/ELF

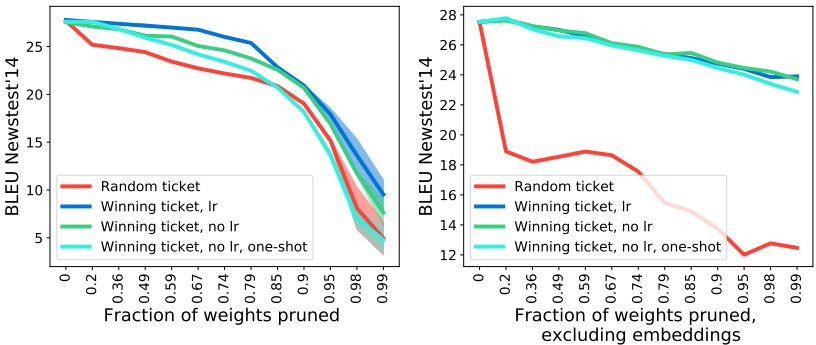

**Figure 2:** Winning ticket initialization performance for Transformer Base models trained on machine translation.

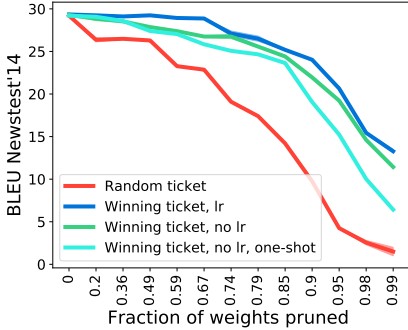

**Figure 3:** Winning ticket initialization performance for Transformer Big models trained on machine translation.

We also evaluated a version of the Transformer Base model in which only Transformer layer weights (attention and fully connected layers) were pruned, but embeddings were left intact (Figure 2 right). Results in this setting were noticeably different from when we pruned all weights. First, the random ticket performance drops off at a much faster rate than in the full pruning setting. This suggests that, for random initializations, a large fraction of embedding weights can be pruned without damaging network performance, but very few transformer layer weights can be pruned. Second, and in stark contrast to the random ticket case, we observed that winning ticket performance was remarkably robust to pruning of only the transformer layer weights, with a roughly linear drop in BLEU score.

To determine whether the lottery phenomenon scales even further, we trained a Transformer Big model with 210M parameters, which reaches a BLEU score of 29.3 (compared to 28.4 in Vaswani et al. (2017)). Here, we again observe that winning tickets significantly outperform random tickets (Figure 2). Notably, we found that with iterative pruning and late rewinding model performance we can train models from scratch 99% of the unpruned model's performance with only a third of the weights (28.9 BLEU 67% pruned vs. 29.2 BLEU unpruned). This result demonstrates the practical implications for the lottery ticket phenomenon as current state of the art Transformer-based systems are often too large to deploy and are highly expensive to train, though sparse matrix multiplication libraries would be necessary to realize this gain.

## 4.2 REINFORCEMENT LEARNING

### 4.2.1 CLASSIC CONTROL

To begin our investigation of the lottery ticket hypothesis in reinforcement learning, we evaluated three simple classic control games: Cartpole-v0, Acrobot-v1, and LunarLander-v2. As discussed in Section 3.3.1 and Table A1, we used simple fully-connected models with three hidden layers. Encouragingly, and consistent with supervised image classification results, we found that winning tickets successfully outperformed random tickets in classic control environments (Figure 4). This

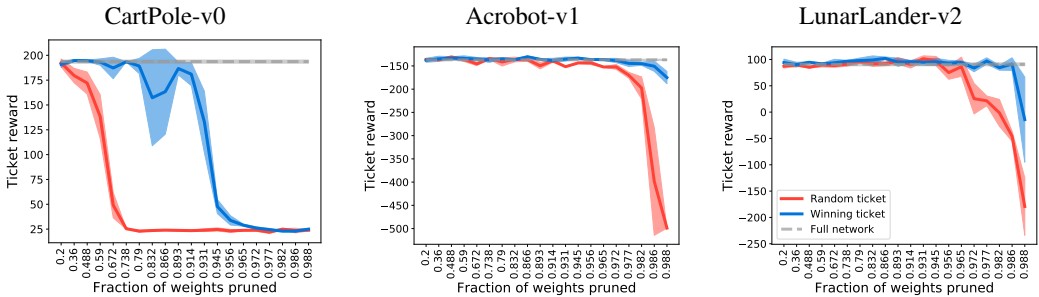

**Figure 4:** Winning ticket performance on classic control games. Each curve plots the mean $\pm$ standard deviation of three independent iterative pruning processes for each game.

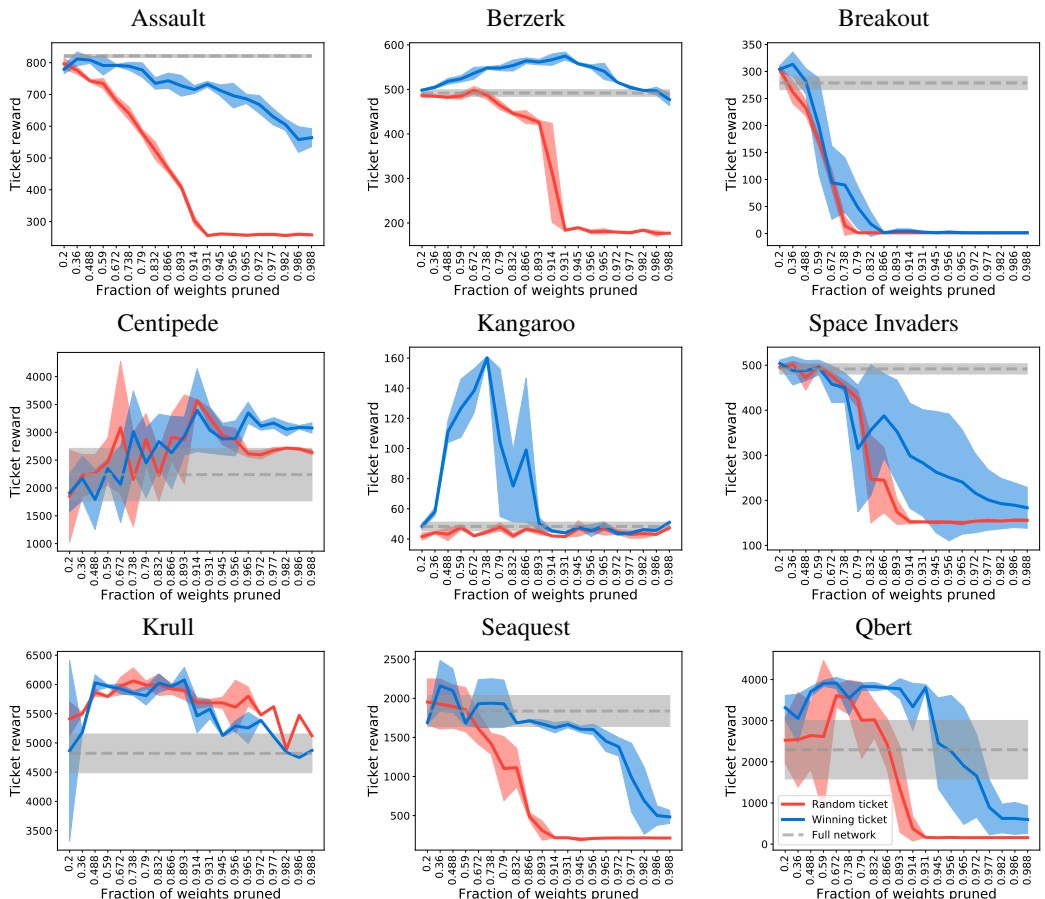

**Figure 5:** Reward curves of WTs (blue) and RTs (red) on Atari. Shaded error bars represent mean $\pm$ standard deviation across runs and the gray curve represents performance of the unpruned network.

result suggests that the lottery ticket phenomenon is not merely an artifact of supervised image classification, but occurs in RL paradigms as well.

### 4.2.2 ATARI GAMES

However, in the original lottery ticket study (Frankle & Carbin, 2019), winning tickets were substantially easier to find in simple, fully-connected models trained on simple datasets (e.g., MNIST) than in more complex models trained on larger datasets (e.g. ResNets on CIFAR and ImageNet). We therefore asked whether winning tickets exist in convolutional networks trained on Atari games as

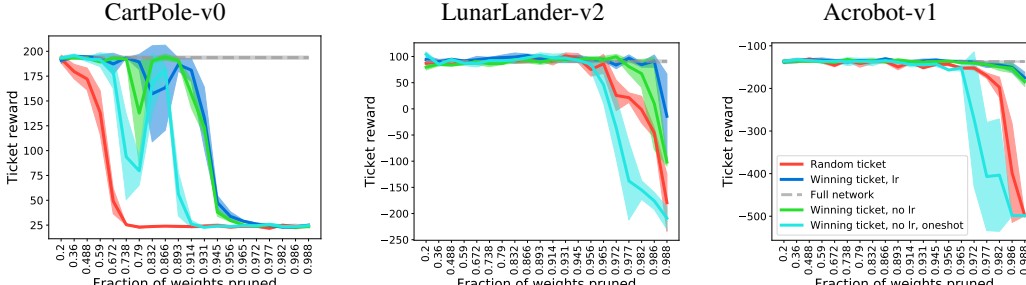

**Figure 6:** Ablation studies of several classic control games on the effects of late rewinding and iterative pruning. Shaded error bars represent mean ± standard deviation across runs and the gray curve represents performance of the unpruned network.

well. We found that the impact of winning tickets varied substantially across Atari games (Figure 5), with some games, such as Assault, Seaquest, and Berzerk benefiting significantly from winning ticket initializations, while other games, such as Breakout and Centipede only benefitted slightly. Notably, winning ticket initializations *increased* reward for both Berzerk and Qbert. Interestingly, one game, Krull, saw no such benefit, and both winning and random tickets performed well even at the most extreme pruning fractions, suggesting that Krull may be so over-parameterized that we were unable to get into the regime in which winning ticket differences emerge.

One particularly interesting case is that of Kangaroo. Because we used the same hyperparameter settings for all games, the initial, unpruned Kangaroo models failed to converge to high rewards (typical reward on Kangaroo for converged models is in the several thousands). Surprisingly, however, winning ticket initializations substantially improved performance (though these models were still very far from optimal performance on this task) over random tickets, enabling some learning where no learning at all was previously possible. Together, these results suggest that while beneficial winning ticket initializations can be found for some Atari games, winning ticket initializations for other games may not exist or be more difficult to find.

We also observed that the shape of pruning curves for random tickets on Atari games also varied substantially based on the game. For example, some games, such as Breakout and Space Invaders, were extremely sensitive to pruning, with performance dropping almost immediately, while other games, such as Berzerk, Centipede, and Krull actually saw performance steadily *increase* in early pruning iterations. This result suggests that the level of over-parameterization varies dramatically across Atari games and that "one size fits all" models may have subtle impacts on performance based on their level of over-parameterization.

To measure the impact of late rewinding and iterative pruning on the performance of winning ticket initializations in RL, we performed ablation studies on six representative games both from classic control and Atari: CartPole-v0, Acrobot-v1, LunarLander-v2, Assault, Breakout, and Seaquest. For all ablation experiments, we leave all training parameters fixed (configuration, hyperparameter, optimizer, etc.) except for those specified. For both classic control (Figure 6) and Atari (Figure 7), we observed that, consistent with previous results in supervised image classification (Frankle et al., 2019; Frankle & Carbin, 2019), both late rewinding and iterative pruning improve winning ticket performance, though interestingly, the degree to which these modifications improved performance varied significantly across games.

## 5 CONCLUSION

In this study, we investigated whether the lottery ticket hypothesis holds in regimes beyond simple supervised image classification by analyzing both NLP and RL domains. For NLP, we found that winning ticket initializations beat random tickets both for recurrent LSTM models trained on language modeling and Transformer models trained on machine translation. Notably, we found high performing Transformer Big models even at high pruning rates ($\geq 67\%$). For RL, we found that winning ticket initializations substantially outperformed random tickets on classic control problems and for many, but not all, Atari games. Together, these results suggest that the lottery ticket phenomenon is not

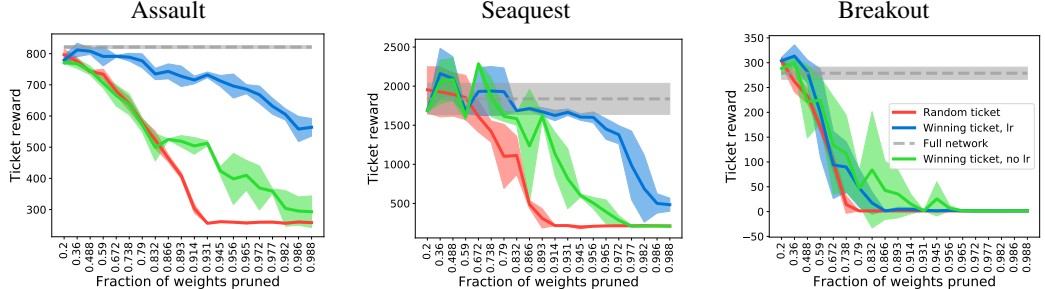

**Figure 7:** Ablation studies of several pixel control games on the effects of iterative pruning. Shaded error bars represent mean ± standard deviation across runs and the gray curve represents performance of the unpruned network. "lr" means late-resetting.

restricted to supervised image classification, but rather represents a general feature of deep neural network training.

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

# A    APPENDIX

| Type | Name | Network specs | Algorithm | $N$ | $M$ | $L$ |
|------|------|---------------|-----------|-----|-----|-----|
| Classic | CartPole-v0 | MLP(128-128-128-out) | A2C | 20 | 160 (games) | 100 |
| | Acrobot-v1 | MLP(256-256-256-out) | A2C | 20 | 320 (games) | 100 |
| | LunarLander-v2 | MLP(256-256-256-out) | A2C | 20 | 640 (games) | 100 |
| Pixel | Assault, Berzerk, Breakout, Centipede, Kangaroo, Krull, Qbert, Seaquest, Space Invaders | Conv(5,64,1,2)-MaxPool(2) -Conv(5,64,1,2)-MaxPool(2) -Conv(3,64,1,1)-MaxPool(2) -Conv(3,64,1,1)-MaxPool(2) -MLP(1920-512-512-out) | A2C (with importance factor correction) | 25 | 1000 (batches) | 1024 |

Table A1: A summary of the games in our RL experiments. Conv($w$, $x,y,z$) represents a convolution layer of filter size $w$, channel number $x$, stride $y$, and padding $z$, respectively. All the layer activations are ReLUs. See Sec. 3.3.2 for the meaning of $M$, $N$ and $L$.

