# OpenReview forum: "Playing the lottery with rewards and multiple languages: lottery tickets in RL and NLP"
_ICLR.cc/2020/Conference — Accept (Poster)_

### Official Review · AnonReviewer3 · 2019-10-20
**Official Blind Review #3**

**Rating:** 6

**Review:**

This paper describes an application of lottery ticket hypothesis to NLP and RL problems. An extensive set of experiments on very strong baseline models on language modeling, machine translation and atari games demonstrates that lottery ticket hypothesis is not only present in feed-forward and convolutional nets on image classification tasks as demonstrated originally in papers by Frankle & Carbin, 2019 and Frankle et al 2019. I don't have any major complaints regarding this work and I believe it is well executed.

For NLP problems it would be nice to have additional ablation studies comparing lottery ticket hypothesis pruning with methods like distillation (Hinton et al 2014) and attention pruning (Michel et al 2019 Are Sixteen Heads Really Better than One?).

For RL it is quite interesting that pruning weights sometimes improves the final scores on Atari games like Berzerk, Kangaroo, Krull and Centipede perhaps due to exploration. As a future work it would be interesting to see a way to using lottery ticket hypothesis to guide exploration in RL.

Overall it is well executed paper, although it is mainly an application of existing lottery ticket hypothesis techniques to NLP and RL.

**Experience Assessment:**

I have published in this field for several years.

**Review Assessment: Checking Correctness Of Derivations And Theory:**

I assessed the sensibility of the derivations and theory.

**Review Assessment: Checking Correctness Of Experiments:**

I assessed the sensibility of the experiments.

**Review Assessment: Thoroughness In Paper Reading:**

I read the paper at least twice and used my best judgement in assessing the paper.

---

### Official Review · AnonReviewer2 · 2019-10-23
**Official Blind Review #2**

**Rating:** 3

**Review:**

This paper studies the existence of “winning ticket” initialization in natural language processing (NLP) and reinforcement learning (RL). The lottery ticket hypothesis has been found effective in over-parameterized deep neural networks, which provides better sub-network initialization if not outperforming the original full network. Experiments show that winning ticket initializations generally outperform parameter-matched random initializations on recurrent LSTM models, large-scale Transformer, and discrete-action space RL tasks, including both classic control and pixel control.  A substantial number of parameters can be saved.

This paper is clearly motivated and easy to follow.  The results are interesting.  However, my major concern is its intellectual merit.  The paper does not seem to propose any new algorithm, and the application of lottery ticket (late rewinding) to LSTM, Transformer, and RL looks quite straightforward. There are a few insights drawn from the experiment, such as when only the Transformer weights are pruned.  However they does not appear substantial. Overall, I do not find the paper innovative enough for publication at top conferences like ICLR.  I understand this is subjective, so I leave it to the AC for further evaluation.

**Experience Assessment:**

I do not know much about this area.

**Review Assessment: Checking Correctness Of Derivations And Theory:**

N/A

**Review Assessment: Checking Correctness Of Experiments:**

I assessed the sensibility of the experiments.

**Review Assessment: Thoroughness In Paper Reading:**

I read the paper at least twice and used my best judgement in assessing the paper.

---

### Official Review · AnonReviewer1 · 2019-11-05
**Official Blind Review #1**

**Rating:** 3

**Review:**

The paper used the lottery ticket hypothesis to study the over-parameterization of deep neural networks (DNNs). The main idea is that overparametrization increases the probability of a “lucky” sub-network initialization being present rather than by helping the optimization process.

The paper conducted experiments to evaluate whether “winning ticket” initializations exist in two different domains: natural language processing (NLP) and reinforcement learning (RL). The authors confirm that winning ticket initializations
generally outperform parameter-matched random initializations, even at extreme pruning rates for both NLP and RL. The results suggest that the lottery ticket
hypothesis is not restricted to supervised learning

The similarity between supervised learning and RL and NLP problem is obvious from a function approximation and optimization point of view. The paper is empirical in nature, and do not offer any additional insight.
The experiments are not very conclusive.

**Experience Assessment:**

I have published one or two papers in this area.

**Review Assessment: Checking Correctness Of Derivations And Theory:**

I assessed the sensibility of the derivations and theory.

**Review Assessment: Checking Correctness Of Experiments:**

I assessed the sensibility of the experiments.

**Review Assessment: Thoroughness In Paper Reading:**

I made a quick assessment of this paper.

---

### Author Response · Authors · 2019-11-13
**General response to reviewers**

We thank the reviewers for their time and for their comments. We were pleased to see that the reviewers found the paper “clearly motivated and easy to follow” (R2) and that “overall it is a well executed paper” (R3).

The main issue the reviewers had with this work is primarily that we do not propose any new algorithms, but rather evaluate whether the lottery ticket hypothesis is present in two domains which are substantially different from supervised image classification: RL and NLP.

While we readily acknowledge that our paper does not make an algorithmic contribution, and while novel algorithms are clearly important, we argue that careful and rigorous empirical work to better understand the benefits and limitations of deep learning phenomena (such as the lottery ticket hypothesis) are equally important. Over the last several years, rigorous empirical work has been increasingly sought after in the community, as demonstrated by a number of recent workshops and talks which emphasize this style of work [1-3]. Furthermore, a number of careful empirical studies without algorithmic contributions have been honored recently at ICLR and other conferences with best paper awards [4-5].

Given the impact of the original lottery ticket hypothesis paper [6] (as a rough estimate, it appears by searching on openreview, there are at least 20 submissions to ICLR 2020 related to the lottery ticket hypothesis), we would argue that clearly establishing the robustness of this phenomenon to different learning paradigms and architectures is a valuable and impactful contribution. To this end, we endeavored to evaluate this hypothesis in a diverse set of experiments, including recurrent LSTMs, Transformers, classic control tasks, and Atari tasks. We also performed a detailed set of ablation studies for all of these domains, demonstrating the importance of iterative pruning and rewinding to the discovery of high performance winning tickets.

We strongly disagree with R1’s comment that “the similarity between supervised learning and RL and NLP problem is obvious from a function approximation and optimization point of view”. Note that even in image classification, VGG and ResNet behaves very differently (e.g., Compared to ResNet, VGG is more over-parameterized and more weights can be pruned without affecting performance [8]). The difference is even larger between supervised learning and RL. [9] shows that even with a perfect representation of the state (and thus a good function approximator is easy to construct), it might still take an exponential number of trajectories for an agent to learn good policies. [11] shows that a nonlinear function approximation is still needed for RL to converge to the best solution in the linear function class. And [12] shows that even if the final strategy is linear, a nonlinear function is still needed during the optimization. All these show complicated interactions between the value/policy function update and function approximation [10] in RL. These are all nontrivial phenomena that do not appear in supervised learning.

Furthermore, a number of techniques which are extremely effective in supervised image classification, such as batch normalization, have been much less successful in NLP due to different architectural paradigms and data sources [13]. For example, BERT, one of the most successful NLP models of recent years, does not use these techniques [14]. NLP generation tasks such as machine translation are also substantially different from typical image classification problems. These are structured prediction problem and it has been shown that techniques like beam search have non-trivial impact on generation quality (e.g. even though at training time we optimize the likelihood of individual tokens, at generation time higher likelihood output is not always the best) [15,16].

We note that our results also refute the results of previous studies [7] which investigated the lottery ticket phenomenon in the context of Transformers and were unable to find good winning tickets (critically, this study did not use iterative pruning and late rewinding, which we showed to be essential). We also emphasize that using this approach, we were able to train large Transformer models from scratch to high accuracy while using only one-third the parameters, in stark contrast to previous results.

---

> ### Author Response · Authors · 2019-11-13
> **References for general response to reviewers**
>
> [1] Identifying and Understanding Deep Learning Phenomena, ICML 2019 workshop, http://deep-phenomena.org/
>
> [2] Science meets Engineering of Deep Learning, NeurIPS 2019 workshop, https://sites.google.com/view/sedl-neurips-2019/main
>
> [3] Ali Rahimi, Test of Time award talk, NeurIPS 2017, https://www.youtube.com/watch?v=Qi1Yry33TQE
>
> [4] Understanding deep learning requires rethinking generalization, Chiyuan Zhang, Samy Bengio, Moritz Hardt, Benjamin Recht, and Oriol Vinyals, ICLR 2017, Best Paper Award
>
> [5] Challenging Common Assumptions in the Unsupervised Learning of Disentangled Representations, Francesco Locatello, Stefan Bauer, Mario Lucic, Gunnar Rätsch, Sylvain Gelly, Bernhard Schölkopf, and Olivier Bachem, ICML 2019, Best Paper Award
>
> [6] The Lottery Ticket Hypothesis: Finding Sparse, Trainable Neural Networks, Jonathan Frankle and Michael Carbin, ICLR 2019, Best Paper Award
>
> [7] The State of Sparsity in Deep Neural Networks, Trevor Gale, Erich Elsen, and Sara Hooker, arxiv 2019
>
> [8] Rethinking the Value of Network Pruning. Zhuang Liu, Mingjie Sun, Tinghui Zhou, Gao Huang, Trevor Darrell, ICLR 2019
>
> [9] Is a Good Representation Sufficient for Sample Efficient Reinforcement Learning? Simon S. Du, Sham M. Kakade, Ruosong Wang, Lin F. Yang, arXiv 2019.
>
> [10] R. S. Sutton and A. G. Barto. Reinforcement Learning: An Introduction. The MIT press, Cambridge MA, 2018.
>
> [11] Non-delusional Q-learning and value-iteration, Tyler Lu, Dale Schuurmans, Craig Boutilier, NeurIPS 2018. Best Paper Award
>
> [12] Real-world Video Adaptation with Reinforcement Learning, Hongzi Mao, Shannon Chen, Drew Dimmery, Shaun Singh, Drew Blaisdell, Yuandong Tian, Mohammad Alizadeh, Eytan Bakshy, ICML Workshop 2019
>
> [13] Layer Normalization, Jimmy Lei Ba, Jamie Ryan Kiros, Geoffrey E. Hinton, arxiv 2016
>
> [14] BERT: Pre-training of Deep Bidirectional Transformers for Language Understanding, Jacob Devlin, Ming-Wei Chang, Kenton Lee, Kristina Toutanova, arxiv 2018
>
> [15] On NMT Search Errors and Model Errors: Cat Got Your Tongue?
> Felix Stahlberg, Bill Byrne, EMNLP’19
>
> [16] Analyzing Uncertainty in Neural Machine Translation
> Myle Ott, Michael Auli, David Grangier, Marc'Aurelio Ranzato, ICML’18

---

### Public Comment · ~Shrey_Desai1 · 2019-12-24
**Lottery Tickets in NLP Cite**

Hi, great paper and congratulations on the acceptance! I'm one of the authors of "Evaluating Lottery Tickets Under Distributional Shifts" (https://arxiv.org/abs/1910.12708). For the camera-ready, if possible, please consider citing our paper as we also explore lottery tickets in the natural language processing domain.

---

### Decision · Program_Chairs · 2019-12-19

**Decision:**

Accept (Poster)

**Comment:**

This paper explores the application of the lottery ticket hypothesis to NLP and RL problems for better initialisations of deep networks and reduced model sizes. This is evaluated in a variety of settings, including continuous control and ATARI games for RL, and LSTMs and Transformers for NLP, showing very positive results.

The main issue raised by the reviewers was the lack of algorithmic novelty in the paper. Despite this, I believe the paper to present an important contribution that could stimulate much additional research. The paper is well written and the results are rigorous and interesting. For these reasons I recommend acceptance.